# Association between the shock index on admission and in-hospital mortality in the cardiac intensive care unit

Mitchell Padkins[1], Kianoush Kashani[2,3], Meir Tabi[4], Ognjen Gajic[2], Jacob C. Jentzer[1,2]*

1 Department of Cardiovascular Medicine, Mayo Clinic, Rochester, Minnesota, United States of America, 2 Division of Pulmonary and Critical Care Medicine, Department of Medicine, Mayo Clinic, Rochester, Minnesota, United States of America, 3 Division of Nephrology and Hypertension, Department of Medicine, Mayo Clinic, Rochester, Minnesota, United States of America, 4 Division of Cardiovascular Medicine, Department of Medicine, Jesselson Integrated Heart Center, Jerusalem, Israel

* jentzer.jacob@mayo.edu

## Abstract

### Background

An elevated shock index (SI) predicts worse outcomes in multiple clinical arenas. We aimed to determine whether the SI can aid in mortality risk stratification in unselected cardiac intensive care unit patients.

### Methods

We included admissions to the Mayo Clinic from 2007 to 2015 and stratified them based on admission SI. The primary outcome was in-hospital mortality, and predictors of in-hospital mortality were analyzed using multivariable logistic regression.

### Results

We included 9,939 unique cardiac intensive care unit patients with available data for SI. Patients were grouped by SI as follows: < 0.6, 3,973 (40%); 0.6–0.99, 4,810 (48%); and ≥ 1.0, 1,156 (12%). After multivariable adjustment, both heart rate (adjusted OR 1.06 per 10 beats per minute higher; CI 1.02–1.10; p-value 0.005) and systolic blood pressure (adjusted OR 0.94 per 10 mmHg higher; CI 0.90–0.97; p-value < 0.001) remained associated with higher in-hospital mortality. As SI increased there was an incremental increase in in-hospital mortality (adjusted OR 1.07 per 0.1 beats per minute/mmHg higher, CI 1.04–1.10, p-Value < 0.001). A higher SI was associated with increased mortality across all examined admission diagnoses.

### Conclusion

The SI is a simple and universally available bedside marker that can be used at the time of admission to predict in-hospital mortality in cardiac intensive care unit patients.

**Data Availability Statement:** All relevant data are within the manuscript and its Supporting Information files.

**Funding:** The author(s) received no specific funding for this work.

**Competing interests:** The authors have declared that no competing interests exist.

## Introduction

Hypotension and tachycardia, reflecting hemodynamic instability and associated cardiovascular compensation, are important prognostic markers in acutely ill patients [1, 2]. An elevated shock index (SI), defined as the ratio of heart rate (HR) to systolic blood pressure (SBP), has been shown in multiple clinical arenas to predict worse outcomes. A higher SI has been found to predict hospital admissions and inpatient mortality in an initial triage setting in the Emergency Department (ED) [3, 4]. SI values > 0.9 portend a worse prognosis including increased mortality in patients with sepsis or septic shock and general intensive care unit (ICU) patients [5, 6]. Elevated HR and low SBP are consistent predictors of poor prognosis in patients with acute cardiovascular disease that have been included in many cardiac risk scores, making the SI a promising risk marker [7, 8]. Thus, these easily obtained bedside measurements may be useful for initial triage in an ED or upon ICU admission.

The contemporary cardiac intensive care unit (CICU) cares for a complex mix of patients with acute cardiovascular conditions and critical illness [9–11]. As such, it is logical to assume that the SI would be a prognostic marker in CICU patients, but no prior studies have comprehensively examined the association between SI and outcomes in the CICU. The CICU patient population may be inherently different from other ICU populations due to underlying cardiac rhythm abnormalities that could influence HR and therefore SI [9–11]. Preliminary analyses have shown that CICU patients with a SI ≥1 comprise a high-risk subgroup, but less is known about the utility of the SI for risk-stratification across the spectrum of illness in the CICU. This study aimed to determine whether the SI can aid in prognostication and predict mortality in unselected CICU patients.

## Methods

This retrospective observational cohort study was approved by the Mayo Clinic Institutional Review Board (IRB # 16–000722). The need for informed consent was waived by the Mayo Clinic IRB due to the minimal risk nature of the study under the Declaration of Helsinki. This was a historical cohort analysis using a secondary analysis of a previously constructed institutional database of patients admitted to the CICU at the Mayo Clinic Hospital, St. Mary's Campus. According to Minnesota state law statute 144.295, patients may decline to have their medical records utilized for observational research; patients who did not provide written or verbal Minnesota Research Authorization were therefore excluded from the study.

We retrospectively analyzed data from the index CICU admission of consecutive unique adult patients aged ≥ 18 years admitted to the CICU at Mayo Clinic Hospital St. Mary's Campus between January 1, 2007, and December 31, 2015 who had HR and SBP documented on CICU admission; patients without data for these variables were excluded. The data was then accessed and analyzed April 2022 through February 2023. The SI was calculated using the admission values of HR and SBP [12–21]. Patients were divided into SI groups based on SI cut-offs reported in prior s [22].

We electronically extracted demographic, vital signs, laboratory, clinical and outcome data, procedures, and therapies performed during the index CICU and hospital stay [12–21]. The admission vital signs, clinical measurements, and laboratory values were defined as the first value recorded after or closest to the index CICU admission. Admission diagnoses were defined as all International Classification of Diseases (ICD)-9 diagnostic codes within one day before or after the index CICU admission; these were not mutually exclusive [14, 16–18, 21].

The Acute Physiology and Chronic Health Evaluation (APACHE)-III score, APACHE-IV predicted hospital mortality, and SOFA scores were automatically calculated using data from the first 24 hours of CICU admission using previously validated electronic algorithms [12, 13, 15, 17, 23]. Organ failure was defined as a score ≥ 3 on any of the non-cardiovascular Sequential Organ Failure Assessment (SOFA) organ sub-scores on CICU day 1 [17]. The Mayo Clinic

CICU Admission Risk Score (M-CARS) was calculated on admission as previously described [21]. Individual co-morbidities used to calculate the Charlson Comorbidity Index (CCI) were extracted from the medical record using a previously validated electronic algorithm [24]. The Braden Skin Score was used as a surrogate marker of frailty [15, 21, 25]. Acute kidney injury (AKI) during the CICU stay was identified and staged using creatinine based modified KDIGO criteria, as previously described; severe AKI was defined as KDIGO stage 2 or 3 AKI [26, 27]. To determine the SCAI Shock stage, we first adjudicated hypotension/tachycardia, hypoperfusion, deterioration, and refractory shock using data from CICU admission through the first 24 hours in the CICU (**S1 Table**) [14, 16, 19, 20, 28]. We mapped the five SCAI Shock stages of increasing severity (A through E) using combinations of these variables (**S2 Table**) [14, 16, 18, 20, 28, 29]. The Vasoactive-Inotropic Score (VIS) was calculated using peak vasopressor and inotrope doses [30].

All-cause CICU and in-hospital mortality were determined using an electronic review of medical records for notification of patient death. Categorical variables are reported as numbers (percentage), and the Pearson chi-square test was used to compare groups. Continuous variables are reported as median and interquartile range (IQR), and the Kruskal-Wallis rank-sum test was used to compare groups. Discrimination was assessed using the area under the receiver-operator characteristic curve (AUC, C-statistic) values. Odds ratio (OR) and 95% confidence interval (CI) values for in-hospital mortality were estimated using logistic regression before and after multivariable adjustment. Multivariable models were adjusted for the following variables selected *a priori* based on clinical relevance and known associations with in-hospital mortality in CICU patients: age, gender, invasive ventilation, CCI, SOFA score, peak VIS, dialysis, percutaneous coronary intervention (PCI), angiogram, cardiac arrest, shock, respiratory failure, Braden skin score, and in-hospital arrest; models were run both with and without SCAI shock stage as an additional variable (**S3 Table**). All P values were two-tailed. Statistical analyses were performed using BlueSky version 7.4 (BlueSky LLC, Chicago, IL).

## Results

Among 10,004 unique CICU patients in our database, we included 9,939 with available data for SI. Median HR, SBP, and shock index for our cohort was 79 (67, 94) beats per minute (BPM), 121 (105, 139) mmHg, and 0.65 (0.63, 0.68) BPM/mmHg, respectively. Patients were grouped by SI: <0.6, 3,973 (40%) patients; 0.6–0.99, 4,810 (48%) patients; and ≥ 1.0, 1,156 (12%) patients (**Fig 1**). The baseline characteristics of these groups are outlined in **Table 1**. The median age was 69 (57, 79) years old, and 37% were females. Admission diagnoses included cardiac arrest in 1,180 (12%), shock in 1,334 (14%, of whom 1,064 [79.8%] had cardiogenic shock), congestive heart failure in 4,545 (46%), acute coronary syndrome in 4,231 (43%), arrhythmias in 4,799 (48%), and sepsis in 602 (6%). Mechanical support devices, pulmonary artery catheter, and vasoactive medications were used in 931 (9%), 721 (7%), and 2,452 (25%) respectively. The distribution of SCAI shock stages was: A, 4,561 (46%); B, 2,988 (30%); C, 1,566 (16%); D, 729 (7%); and E, 95 (1%). A total of 886 (9%) died during the hospitalization, including 551 (5.5%) during the CICU stay; 223 (2.2%) died within 24 hours of CICU admission. The median SI was higher for in-hospital deaths than hospital survivors (0.80 versus 0.64, p <0.001), reflecting lower median SBP (109 versus 122 mmHg) and higher median HR (88 versus 78 BPM, p <0.001) in the deceased group.

### Heart rate and systolic blood pressure

The first recorded HR had a strong positive association with in-hospital mortality (unadjusted OR 1.14 per 10 BPM higher; CI 1.11–1.17; p-value < 0.001; **Fig 2A**) and the first measured

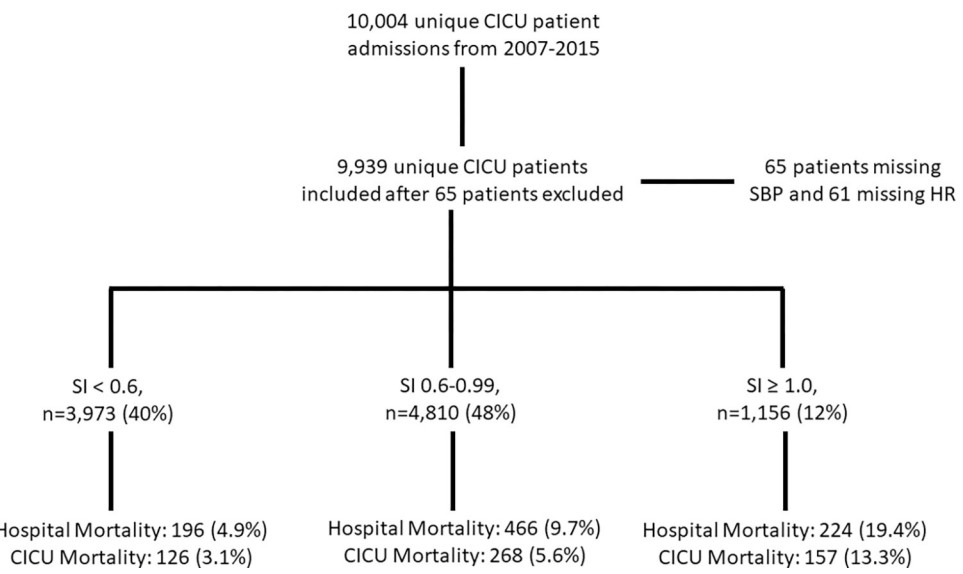

**Fig 1. Consort diagram demonstrating number of patients in each SI group and overall outcomes.**

SBP had a strong inverse association with in-hospital mortality (unadjusted OR 0.84 per 10 mmHg; CI 0.81–0.86; p-value < 0.001; **Fig 2B**). When HR and SBP were combined, a gradient of CICU and in-hospital mortality was observed with increasing HR or decreasing SBP (**Fig 3A**). The lowest CICU and in-hospital mortality was in the group with SBP ≥ 90 mmHg and HR < 100 BPM, while the highest CICU and in-hospital mortality was in the group with SBP < 90 mmHg and HR ≥ 100 BPM (**Fig 3B**).

After multivariable adjustment, both HR (adjusted OR 1.06 per 10 BPM higher; CI 1.02–1.10; p-value 0.005) and SBP (adjusted OR 0.94 per 10 mmHg higher; CI 0.90–0.97; p-value < 0.001) remained associated with higher in-hospital mortality when considered individually. Similar findings were observed when these variables were included together in a multivariable model, HR (adjusted OR 1.06 per 10 BPM higher; CI 1.02–1.10) and SBP (adjusted OR 0.94 per 10 mmHg higher; CI 0.90–0.97; p-value < 0.001).

## Shock index

As SI increased, there was a progressive and incremental increase in in-hospital mortality (unadjusted OR 1.19 per 0.1 BPM/mmHg higher, CI 1.16–1.21, p < 0.001; **Fig 4**). A higher SI (particularly SI ≥1) was associated with increased in-hospital mortality across different admission diagnoses (**Table 2**, **Fig 5**) and in patients supported with invasive ventilation, vasoactive drugs, or mechanical circulatory support (**S1 Fig**). The association between SI and in-hospital mortality was strongest in patients with acute coronary syndromes (ACS) and weaker in patients with critical care diagnoses (**Table 3**). A higher SI was associated with increased in-hospital mortality across the SCAI shock stages, except for SCAI Shock Stage E (**Table 2**, **S2 Fig**). Increasing SI remained associated with higher in-hospital mortality in patients stratified by SBP and HR (**S3 Fig**). Additionally, a higher SI was associated with increased in-hospital mortality when patients were stratified by quartiles of age, Day 1 SOFA score, or APACHE-IV predicted hospital mortality (**S4 Fig**). Notably, the AUC for the SI alone was minimally higher than that for SBP and HR combined.

After multivariable adjustment, a higher SI remained incrementally associated with increased in-hospital mortality (adjusted OR 1.07 per 0.1 BPM/mmHg higher, CI 1.04–1.10,

**Table 1.**

| Variable | Shock Index | | | | |
|---|---|---|---|---|---|
| *Demographics* | < 0.6 (n = 3973) | 0.6–0.99 (n = 4810) | ≥ 1.0 (n = 1156) | Total (n = 9939) | p-Value |
| Age | 71.2 (60.3, 80.4) | 67.5 (55.7, 77.7) | 67.8 (55.1, 77.4) | 69.1 (57.8, 78.9) | < 0.001 |
| Female | 1424 (35.8%) | 1846 (38.4%) | 451 (39%) | 3721 (37.4%) | 0.025 |
| BMI (kg/m²) | 28.5 (25.1, 32.8) | 28.5 (25, 33.3) | 28 (23.9, 32.8) | 28.4 (24.9, 33.1) | 0.015 |
| CCI | 1 (0, 3) | 2 (0, 4) | 2 (0, 4) | 2 (0, 4) | < 0.001 |
| Hospital Length of Stay (Days) | 3.8 (2.3, 6.8) | 5.1 (2.9, 10) | 6.1 (3.1, 11.6) | 4.6 (2.7, 9) | < 0.001 |
| ICU Length of Stay (Days) | 1.6 (0.9, 2.6) | 1.8 (1, 3) | 1.9 (0.9, 3.6) | 1.7 (0.9, 2.9) | < 0.001 |
| Hospital Mortality | 196 (4.9%) | 466 (9.7%) | 224 (19.4%) | 886 (8.9%) | < 0.001 |
| CICU Mortality | 126 (3.1%) | 268 (5.6%) | 157 (13.3%) | 551 (5.5%) | < 0.001 |
| Left Ventricular Ejection Fraction | 55 (43, 62) | 47 (30, 60) | 41 (25, 60) | 50 (35, 60) | < 0.001 |
| *Comorbidities* | | | | | |
| MI | 791 (19.9%) | 959 (20%) | 217 (18.8%) | 1967 (19.8%) | 0.641 |
| CHF | 610 (15.4%) | 1056 (22%) | 282 (24.4%) | 1948 (19.7%) | < 0.001 |
| CVA | 514 (13%) | 578 (12.1%) | 131 (11.4%) | 1223 (12.3%) | 0.244 |
| Moderate/Severe Kidney Disease | 781 (19.7%) | 991 (20.7%) | 247 (21.4%) | 2019 (20.4%) | 0.344 |
| Dialysis | 1121 (28.3%) | 1394 (29.1%) | 303 (26.3%) | 2818 (28.4%) | 0.156 |
| Diabetes Mellitus | 1121 (28.3%) | 1394 (29.1%) | 303 (26.3%) | 2818 (28.4%) | 0.156 |
| Cancer | 802 (20.2%) | 1041 (21.7%) | 280 (24.3%) | 2123 (21.4%) | 0.01 |
| Lung Disease | 661 (16.7%) | 1009 (21%) | 260 (22.5%) | 1930 (19.5%) | < 0.001 |
| Liver Disease | 53 (1.3%) | 113 (2.4%) | 29 (2.5%) | 195 (2%) | 0.001 |
| *Admission Diagnoses* | | | | | |
| Cardiac Arrest | 392 (10%) | 598 (12.5%) | 190 (16.6%) | 1180 (12%) | < 0.001 |
| Shock | 243 (6.2%) | 722 (15.2%) | 369 (32.2%) | 1334 (13.6%) | < 0.001 |
| Sepsis | 94 (2.4%) | 329 (6.9%) | 179 (15.6%) | 602 (6.1%) | < 0.001 |
| Respiratory Failure | 539 (13.7%) | 1134 (23.8%) | 400 (34.9%) | 2073 (21.1%) | < 0.001 |
| Congestive Heart Failure | 1364 (34.8%) | 2484 (52.1%) | 697 (60.9%) | 4545 (46.2%) | < 0.001 |
| Acute Coronary Syndrome | 1782 (45.4%) | 2051 (43%) | 398 (34.8%) | 4231 (43%) | < 0.001 |
| AF/SVT | 1050 (26.8%) | 1605 (33.7%) | 550 (48%) | 3205 (32.6%) | < 0.001 |
| VT/VF | 516 (13.1%) | 852 (17.9%) | 226 (19.7%) | 1594 (16.2%) | < 0.001 |
| Cardiogenic Shock | 198 (5%) | 572 (12%) | 294 (25.7%) | 1064 (10.8%) | < 0.001 |
| **Severity of Illness** | | | | | |
| Apache 3 Score | 55 (42, 69) | 58 (44, 74) | 70 (55, 89) | 58 (44, 73) | < 0.001 |
| Apache 4 Predicted Mortality | 0.1 (0, 0.2) | 0.1 (0, 0.2) | 0.2 (0.1, 0.4) | 0.1 (0, 0.2) | < 0.001 |
| SOFA | 2 (1, 4) | 3 (1, 5) | 4 (2, 8) | 2 (1, 5) | < 0.001 |
| Braden Skin Score | 18 (16, 20) | 18 (15, 20) | 17 (14, 19) | 18 (15, 20) | < 0.001 |
| Systolic Blood Pressure | 138 (123, 154) | 114 (102, 128) | 95 (84, 107) | 121 (105, 139) | < 0.001 |
| Heart Rate | 66 (58, 74) | 86 (76, 97) | 114 (100, 129) | 79 (67, 94) | < 0.001 |
| First Mean Arterial Pressure | 87 (77, 98) | 80 (71, 90) | 72 (62, 83) | 82 (72, 93) | < 0.001 |
| Oxygen saturation | 98 (95, 99) | 97 (94, 99) | 96 (93, 98) | 97 (94, 99) | < 0.001 |
| M-CARS | 1 (0, 2) | 2 (0, 3) | 3 (1, 5) | 1 (0, 3) | < 0.001 |
| Any AKI in CICU | 1093 (30.4%) | 1848 (42.9%) | 531 (53.7%) | 3472 (39.1%) | < 0.001 |
| Any AKI in Hospital | 1608 (42.9%) | 2453 (54.7%) | 659 (64%) | 4720 (50.9%) | < 0.001 |
| Severe AKI in CICU | 281 (7.8%) | 639 (14.8%) | 209 (21.1%) | 1129 (12.7%) | < 0.001 |
| Severe AKI in Hospital | 398 (10.6%) | 877 (19.5%) | 271 (26.3%) | 1546 (16.7%) | < 0.001 |
| **SCAI Cardiogenic Shock Stage** | | | | | |
| A | 2676 (67.4%) | 1884 (39.2%) | 1 (0.1%) | 4561 (45.9%) | < 0.001 |
| B | 544 (13.7%) | 1735 (36.1%) | 709 (61.3%) | 2988 (30.1%) | < 0.001 |

*(Continued)*

**Table 1.** (Continued)

| Variable | Shock Index | | | | |
|---|---|---|---|---|---|
| *Demographics* | < 0.6 (n = 3973) | 0.6–0.99 (n = 4810) | ≥ 1.0 (n = 1156) | Total (n = 9939) | **p-Value** |
| C | 569 (14.3%) | 782 (16.3%) | 215 (18.6%) | 1566 (15.8%) | < 0.001 |
| D | 173 (4.4%) | 371 (7.7%) | 185 (16%) | 729 (7.3%) | < 0.001 |
| E | 11 (0.3%0 | 38 (0.8%) | 46 (4%) | 95 (1%) | < 0.001 |
| *Procedures and Therapeutics* | | | | | |
| Vasoactive Medications | 570 (14.3%) | 1374 (28.6%) | 508 (43.9%) | 2452 (24.7%) | < 0.001 |
| Vasoactive-Inotropic Score (VIS) | 0 (0, 0) | 0 (0, 2.5) | 0 (0, 12.6) | 0 (0, 0) | < 0. 001 |
| Dialysis | 100 (2.5%) | 274 (5.7%) | 111 (9.6%) | 485 (4.9%) | < 0.001 |
| Temporary Mechanical Support | 254 (6.4%) | 501 (10.4%) | 176 (15.2%) | 931 (9.4%) | < 0.001 |
| Intra-aortic Balloon Pump | 241 (6.1%) | 460 (9.6%) | 162 (14%0 | 863 (8.7%) | < 0.001 |
| Pulmonary Artery Catheter | 147 (3.7%0 | 445 (9.3%0 | 129 (11.2%) | 721 (7.3%) | < 0.001 |
| Coronary Angiogram | 2266 (57%) | 2523 (52.5%) | 454 (39.3%) | 5243 (52.8%) | < 0.001 |
| Percutaneous Coronary Intervention | 1616 (40.7%) | 1539 (32%) | 253 (21.9%) | 3408 (34.3%) | <0.001 |
| *Laboratory Values* | | | | | |
| Hemoglobin (g/dL) | 12.5 (11, 13.9) | 12 (10.5, 13.6) | 11.6 (10.1, 13.3) | 12.2 (10.6, 13.7) | < 0.001 |
| Platelets (x10$^9$/L) | 200 (160, 244) | 203 (158, 257) | 202 (151.5, 262) | 201 (158, 251) | 0.08 |
| Sodium (mmol/L) | 139 (136, 141) | 138 (135, 140) | 138 (134, 140) | 138 (136, 141) | < 0.001 |
| Potassium (mmol/L) | 4.2 (3.9, 4.6) | 4.2 (3.8, 4.6) | 4.3 (3.9, 4.7) | 4.2 (3.9, 4.6) | < 0.001 |
| Bicarbonate (mmol/L) | 24 (22, 26) | 24 (21, 26) | 23 (20, 26) | 24 (21, 26) | < 0.001 |
| Chloride (mmol/L) | 104 (101, 106) | 103 (99, 106) | 103 (99, 106) | 103 (100, 106) | < 0.001 |
| BUN (mg/dL) | 19 (14, 29) | 21 (15, 33) | 24 (16, 39) | 21 (15, 32) | < 0.001 |
| Creatinine (mg/dL) | 1 (0.8, 1.4) | 1.1 (0.8, 1.4) | 1.2 (0.9, 1.7) | 1 (0.8, 1.4) | < 0.001 |
| eGFR (CKD-EPI) (mg/min/BSA) | 70.8 (46.6, 89.1) | 68.1 (43.6, 89.3) | 59.7 (35.2, 87.1) | 68.5 (43.7, 89.1) | < 0.001 |
| Anion Gap | 11 (9, 13) | 11 (9, 14) | 12 (10, 15) | 11 (9, 14) | < 0.001 |
| Albumin (g/dL) | 3.6 (3.1, 3.9) | 3.4 (2.9, 3.8) | 3.1 (2.7, 3.5) | 3.4 (2.9, 3.8) | < 0.001 |
| Alanine aminotransferase (U/L) | 29 (19, 50) | 33 (20, 64) | 39 (22, 89) | 32 (20, 60) | < 0.001 |
| Aspartate aminotransferase (U/L) | 40 (26, 91) | 47 (29, 110) | 53 (31, 143) | 45 (28, 105) | < 0.001 |
| Lactate (mmol/L) | 1.6 (1, 2.8) | 1.7 (1.1, 2.9) | 2 (1.3, 3.8) | 1.8 (1.1, 3) | < 0.001 |

p-Value < 0.001). In addition, an SI ≥ 1 was associated with higher in-hospital mortality (adjusted OR 1.35, CI 1.07–1.69, p-Value 0.011). Finally, when the SCAI shock stage was included in the model, SI remained directly associated with in-hospital mortality (adjusted OR 1.05 per 0.1 BPM/mmHg higher, CI 1.02–1.08, p-Value 0.002), while SI ≥1 no longer had a significant association (adjusted OR 1.21, CI 0.96–1.52, p-Value 0.11).

## Discussion

This analysis of 9,939 patients demonstrates that SI at the time of CICU admission can be used to add additional prognostic information above and beyond traditional mortality risk factors, including composite severity of illness risk scores. This current analysis is consistent with prior studies in other populations, expanding the prognostic information provided by the SI. Our study extends the known association between SI and mortality to patients with cardiac conditions, including cardiogenic shock and cardiac arrhythmias [5]. The SI can be easily calculated by clinicians at the bedside or automated using the electronic health record to identify high-risk individuals, such as those with a SI ≥1.

Prior studies have demonstrated the prognostic utility of SI in patients with individual cardiac pathologies, including ACS, heart failure, and cardiogenic shock [31–35]. SI alone was

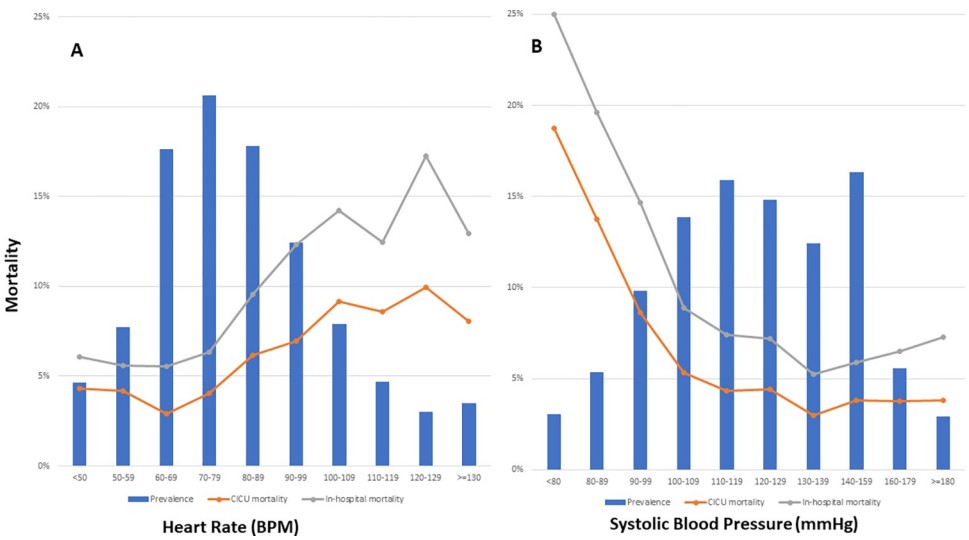

**Fig 2.** A. Bar graph and line graph demonstrating the relationship of HR on mortality. B. Bar graph and line graph demonstrating the relationship of SBP on mortality.

shown to perform as well as the GRACE risk score in patients with ACS, with the advantage of being easier to calculate [36–38]. Notably, both HR and SBP are included in the GRACE risk score [7, 39]. The TIMI Risk Index is calculated as $age^2$ multiplied by the SI and has been shown to predict mortality in ACS and cardiogenic shock [36–38]. Surprisingly, the SI alone was found to outperform the TIMI Risk Score in some studies of ACS patients [34, 35]. Together, SI and lactate concentration can predict 24-hour and 28-day mortality in acute heart failure patients [40]. Further, the ACTION ICU score is another scoring system that utilized HR and SBP and can be used in non-ST segment elevation myocardial infarction patients to predict ICU level of care needs [41]. SI has also been studied in hypovolemic trauma patients and during hemodynamic resuscitation and has been found to predict mortality in these

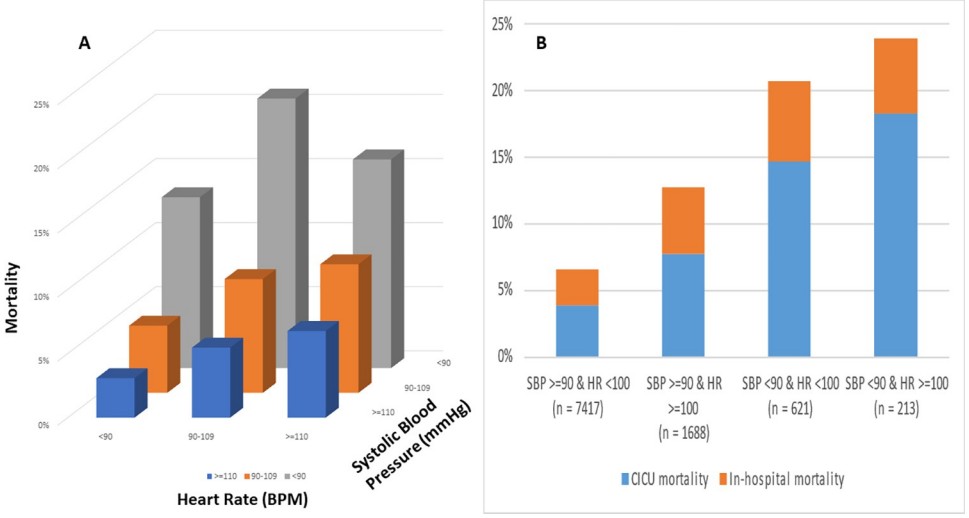

**Fig 3.** A: Bar graph demonstrating the relationship of HR and SBP on mortality. B: Bar graph demonstrating mortality stratified by HR and SBP.

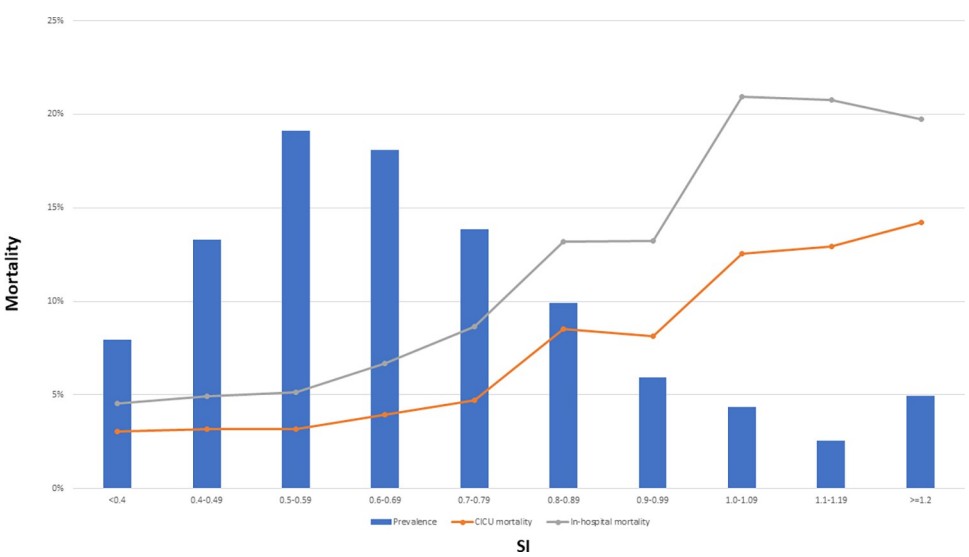

**Fig 4. Line graph demonstrating the relationship between SI and mortality.**

groups as well [42]. All of these studies integrate the HR and SBP to predict risk and prognosticate outcomes, and thus, SI has been shown to predict prognosis and mortality in many patient populations, especially cardiac patients. Our study adds to this evidence by demonstrating its predictive effectiveness in an unselected CICU cohort spanning the spectrum of illness severity and cardiac pathology. Notably, SI provided better discrimination in our ACS patients, who generally had a wider spectrum of risk and a lower baseline mortality rate.

**Table 2.**

| Variable | SI < 0.6 | SI 0.6–0.99 | SI ≥ 1.0 | Total | p-Value |
|---|---|---|---|---|---|
| **Hospital Mortality Overall** | 196 (4.9%) | 466 (9.7%) | 224 (19.4%) | 886 (8.9%) | < 0.001 |
| **Admission Diagnoses** | | | | | |
| **Cardiac Arrest** | 109 (27.8%) | 191 (31.9%) | 96 (50.5%) | 396 (33.6%) | < 0.001 |
| **Shock** | 76 (31.3%) | 226 (31.3%) | 146 (39.6%) | 448 (33.6%) | 0.017 |
| **Cardiogenic Shock** | 64 (32.3%) | 175 (30.6%) | 121 (41.2%) | 360 (33.8%) | 0.007 |
| **Congestive Heart Failure** | 100 (7.3%) | 306 (12.3%) | 139 (19.9%) | 545 (12%) | < 0.001 |
| **Acute Coronary Syndrome** | 76 (4.3%) | 200 (9.8%) | 93 (23.4%) | 369 (8.7%) | < 0.001 |
| **Respiratory Failure** | 118 (21.9%) | 275 (24.3%) | 138 (34.5%) | 531 (25.6%) | < 0.001 |
| **AF/SVT** | 65 (6.2%) | 188 (11.7%) | 87 (15.8%) | 340 (10.6%) | < 0.001 |
| **VT/VF** | 43 (8.3%) | 98 (11.5%) | 43 (19%) | 184 (11.5%) | < 0.001 |
| **Sepsis** | 23 (24.5%) | 94 (28.6%) | 57 (31.8%) | 174 (28.9%) | 0.434 |
| **Mechanical Circulatory Support** | 27 (10.6%) | 91 (18.2%) | 49 (27.8%) | 167 (17.9%) | < 0.001 |
| **Vasoactive Medication** | 103 (18.1%) | 315 (22.9%) | 169 (33.3%) | 587 (23.9%) | < 0.001 |
| **Invasive Ventilation** | 112 (26%) | 241 (28.1%) | 127 (40.3%) | 480 (29.9%) | < 0.001 |
| **SCAI Shock Stage** | | | | | |
| **A** | 60 (2.2%) | 69 (3.7%) | 0 (0%) | 129 (2.8%) | 0.017 |
| **B** | 23 (4.2%) | 118 (6.8%) | 65 (9.2%) | 206 (2.8%) | 0.003 |
| **C** | 46 (8.1%) | 103 (13.2%) | 46 (21.4%) | 195 (12.5%) | < 0.001 |
| **D** | 59 (34.1%) | 148 (39.9%) | 86 (46.5%) | 293 (40.2%) | 0.057 |
| **E** | 8 (72.7%) | 28 (73.7%) | 27 (58.7%) | 63 (66.3%) | 0.313 |

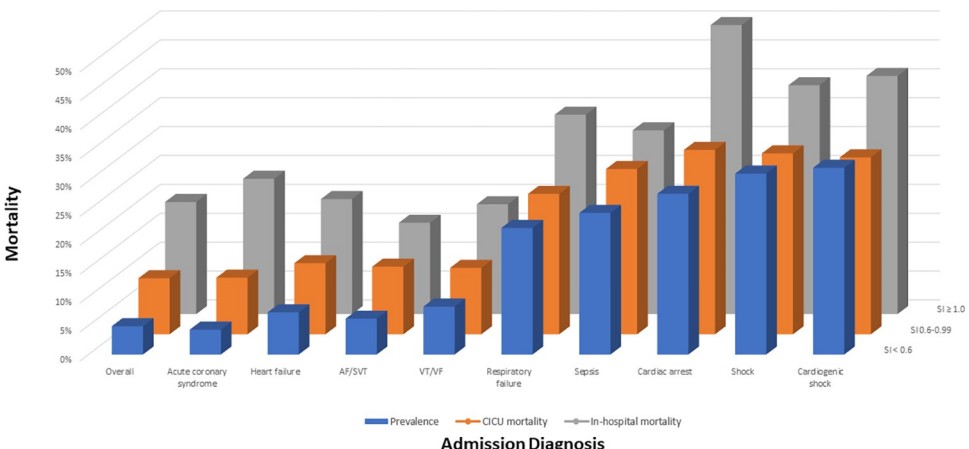

**Fig 5. Bar graph demonstrating the relationship between admission diagnosis, SI, and mortality.**

Generating the SI from ubiquitous vital sign measurements can help risk stratification of patients regardless of the admission diagnoses, SCAI shock stage, or severity of illness. There was a stepwise increase in mortality with higher SI, and patients with a SI ≥1 had a particularly high risk of mortality. However, this effect was mitigated when adjusting for the severity of shock using the SCAI shock stage, likely because SBP, HR, and SI were all used to define hemodynamic instability for the SCAI Shock stages [16]. Among the various metrics used to gauge the presence of hypotension, an elevated SI was most strongly associated with mortality in our cohort [31]. Additionally, tachycardic patients were at higher risk of mortality across the preshock and shock spectrum [31]. Therefore, the SI provides a single metric that may be more useful clinically than HR or SBP alone, or at least provides complementary information. Prior studies have demonstrated the usefulness of adding other risk factors when stratifying patients using the SCAI shock stages and prognostication [16, 19]. This analysis adds SI to the list of potential "risk modifiers" that can refine mortality risk stratification using the SCAI Shock Classification, particularly at lower shock severity. We found that treating SI as a continuous variable had a stronger association with mortality than using a prespecified SI cut-off ≥1, highlighting the incremental prognostic information across the SI range. Interestingly, there was a steep rise in mortality once the SI was ≥0.9, consistent with prior reports of worsening mortality with SI values ≥ 0.9 in ED and ICU settings [3, 4].

**Table 3.**

| Admission Diagnosis | Unadjusted OR per 0.1 unit SI | CI | p-Value | AUC |
|---|---|---|---|---|
| Acute Coronary Syndrome | 1.24 | 1.2–1.28 | < 0.001 | 0.68 |
| AF/SVT | 1.12 | 1.08–1.15 | < 0.001 | 0.62 |
| Cardiac Arrest | 1.13 | 1.08–1.18 | < 0.001 | 0.59 |
| Cardiogenic Shock | 1.07 | 1.03–1.11 | 0.002 | 0.55 |
| Congestive Heart Failure | 1.14 | 1.11–1.18 | < 0.001 | 0.62 |
| Respiratory Failure | 1.09 | 1.06–1.13 | < 0.001 | 0.57 |
| Sepsis | 1.04 | 0.98–1.1 | 0.18 | 0.54 |
| Shock | 1.05 | 1.02–1.09 | 0.005 | 0.55 |
| VT/VF | 1.14 | 1.08–1.2 | < 0.001 | 0.61 |

A high SI can be driven by a low SBP and/or a high HR, either of which can be prognostic in acute illness and typically reflect physiologic stress or hemodynamic compromise. Conversely, an inappropriate HR response to hypotension can result in a relatively lower SI. We have previously observed that the average HR increases in parallel with increasing shock severity in our CICU cohort, as does the SI [2, 16, 31]. Increasing HR may be associated with a worse prognosis in CICU patients because of increased myocardial oxygen demand inducing myocardial ischemia or alternatively may simply be a marker of the severity of the stress response with or without further exacerbation by tachyarrhythmias. Further, decreased blood pressure can lead to tissue hypoperfusion and reduced oxygen delivery reflecting hemodynamic compromise from an underlying condition. These hemodynamic perturbations regarding HR and SBP may represent additional stressors to an at-risk CICU population with tenuous clinical status. Further study is needed to better understand the hemodynamic underpinnings of an elevated SI and determine if a low stroke volume is the dominant etiology as expected.

## Strengths and limitations

The primary strength of this analysis is the inclusion of a large cohort (nearly 10,000 unique patients) of CICU patients spanning nearly a decade (2007 to 2015), with extensive data regarding their hospitalization to provide deep characterization of the population. We purposefully included a heterogeneous group patients including the spectrum of illness severity from those with relatively uncomplicated acute cardiovascular disease to those with severe critical illness. This allowed us to examine the performance of the admission SI as a simple marker for risk stratification in patients with different disease processes and degrees of circulatory failure.

Despite these strengths, this study has inherent limitations to single-center retrospective cohort analyses and prevents drawing causal inferences. A key source of potential bias is selection bias, as this single-center study cohort only included those admitted to the CICU at our facility and may not apply equally to patients who are not candidates for CICU admission for a variety of possible reasons. The CICU population at Mayo Clinic may differ from other populations in terms of baseline demographics, case mix, and resource utilization. Further, this CICU cohort at Mayo Clinic was relatively unselected, and these results may not apply equally to all subsets of the CICU patient population. Residual confounding is always possible in studies such as this one, and in this case an important potential source of residual confounding is overall severity of illness, which increased progressively as the SI increased, and might be incompletely captured with the covariates we used for adjustment. In addition, our analyses focused only on initial vital signs, and it is likely that the evolution of changes in vital signs (e.g., SI) over time could provide enhanced prognostic value; a few patients did not have vital signs documented and were excluded. Further studies would be necessary to determine how a changing SI trajectory over time during a patient's CICU and hospital stay affected mortality risk. Data regarding medication utilization prior to CICU admission (e.g., beta-blockers, anti-arrhythmics, and other antihypertensives) was unavailable and may influence the observed admission HR, SBP, and SI. Notably, other versions of the SI can be calculated, including the modified SI based on the mean arterial pressure and the diastolic SI based on the diastolic blood pressure. We found that the area under the curve value for discrimination of in-hospital mortality was equivalent among these permutations (0.66) and trivially higher for the standard SI. We did not have data regarding the presence of implantable electronic cardiac rhythm devices or anti-arrhythmic drugs, which could have influenced the HR and thus the SI and its association with mortality. Finally, we could not determine each patient's cardiac rhythm

when the SI was calculated, preventing us from determining the effects of tachyarrhythmias and bradyarrhythmias on the prognostic associations of the SI. This is an important potential source of imprecision, as a variety of factors (including antiarrhythmic and vasoactive drugs as well as cardiac rhythm) can independently affect the SI and therefore affect the relationship between SI and outcomes by either weakening this association (i.e., if a change in SI occurred due to a benign condition that did not affect mortality) or artificially augmenting this association (i.e., if an increase in SI occurred due to a harmful condition that worsened outcomes). The true range of SI values within the cohort was very wide, with only a few individuals at the extreme values. Accordingly, we chose to categorize SI values into groups to ensure adequate group size for analysis, but such categorization results in information loss and our estimates are imprecise at the extremes of SI where there were sparse data; this could explain why we did not see dramatic incremental increases in mortality at the highest SI values (i.e., a plateau effect was present to some extent above a SI value of 1).

## Conclusion

Our study is congruous with prior studies clearly demonstrating that a higher SI predicts increased mortality in acutely ill patients. We propose that the SI can be used in the CICU to predict CICU and in-hospital mortality in patients suffering from various cardiac conditions, even after adjustment for standard markers of illness severity. Overall, SI was found to be a better discriminator of mortality than using HR and SBP individually or combined. Further, incremental increases in SI were found to predict mortality when stratified by admission diagnosis and SCAI shock stage. Thus, SI can be used individually or in combination to predict mortality in CICU patients. Further research is necessary to determine if specific patient-level data, such as presenting cardiac arrhythmia, could affect SI and patient outcomes.

## Supporting information

**S1 Fig. Bar graph demonstrating the relationship between mechanical support devices, vasoactive medications, and mechanical ventilation on mortality stratified by SI.**
(TIF)

**S2 Fig. Bar graph demonstrating the relationship between SCAI shock stage and mortality.**
(TIF)

**S3 Fig.** A: Line graph demonstrating the relationship of SI stratified by HR and SBP on CICU mortality. B: Line graph demonstrating the relationship of SI stratified by HR and SBP on hospital mortality.
(TIF)

**S4 Fig. Bar graph demonstrating the relationship of SI stratified by quartiles of age, day 1 SOFA score, and APACHE-IV score on mortality.**
(TIF)

**S1 Table. Study definitions of hypotension, tachycardia, hypoperfusion, deterioration and refractory shock.** Data from Jentzer, et al. J Am Coll Cardiol 2019.
(DOCX)

**S2 Table. Definition of cardiogenic shock (CS) stages used in this study, based on the Society for Cardiovascular Angiography and Intervention (SCAI) consensus statement classification.** Data from Jentzer, et al. J Am Coll Cardiol 2019.
(DOCX)

**S3 Table. Variables used for multi-variable logistic regression model.**
(DOCX)

## Author Contributions

**Conceptualization:** Mitchell Padkins, Kianoush Kashani, Meir Tabi, Ognjen Gajic, Jacob C. Jentzer.

**Investigation:** Mitchell Padkins.

**Methodology:** Mitchell Padkins, Jacob C. Jentzer.

**Resources:** Mitchell Padkins, Kianoush Kashani, Meir Tabi, Ognjen Gajic.

**Supervision:** Jacob C. Jentzer.

**Writing – original draft:** Mitchell Padkins, Jacob C. Jentzer.

**Writing – review & editing:** Mitchell Padkins, Kianoush Kashani, Meir Tabi, Ognjen Gajic, Jacob C. Jentzer.

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
