## [Decision Letter · Decision Letter 0]

11 Dec 2023

PONE-D-23-31759Association between the Shock Index on Admission and In-Hospital Mortality in the Cardiac Intensive Care UnitPLOS ONE

Dear Dr. Padkins,

Thank you for submitting your manuscript to PLOS ONE. After careful consideration, we feel that it has merit but does not fully meet PLOS ONE’s publication criteria as it currently stands. Therefore, we invite you to submit a revised version of the manuscript that addresses the points raised during the review process.

We look forward to receiving your revised manuscript.

Kind regards,

Prof. Gaetano Santulli, MD, PhD, FAHA

Academic Editor

PLOS ONE

Journal Requirements:

Reviewers' comments:

Reviewer's Responses to Questions

**Comments to the Author**

1. Is the manuscript technically sound, and do the data support the conclusions?

Reviewer #1: Yes

Reviewer #2: Yes

Reviewer #3: Yes

2. Has the statistical analysis been performed appropriately and rigorously? 

Reviewer #1: Yes

Reviewer #2: Yes

Reviewer #3: Yes

3. Have the authors made all data underlying the findings in their manuscript fully available?

Reviewer #1: Yes

Reviewer #2: Yes

Reviewer #3: Yes

4. Is the manuscript presented in an intelligible fashion and written in standard English?

Reviewer #1: Yes

Reviewer #2: Yes

Reviewer #3: Yes

5. Review Comments to the Author

Reviewer #1: This study provides compelling evidence regarding the predictive ability of the shock index for mortality in predicting patients in the CICU. However, there are a few suggestions to consider. Firstly, regarding Figure 3A, it appears that the correlation between vital signs and mortality is not sufficiently clear, making it less intuitive for readers to understand the intended message. It is recommended to make moderate adjustments for clarity.

Secondly, Table 3 and Figure 5 seem to convey similar information, differing only in the presentation of shock index as a continuous variable or grouped discussion. I believe that grouping shock index as < 0.6, 0.6-0.99, and ≥ 1.0 may not have significant meaning in this context. Perhaps, it would be more meaningful to analyze shock index as a continuous variable directly. If the author agrees with this idea, I suggest using mortality and non-mortality as grouping criteria for Table 1 and Table 2.

Reviewer #2: Thank you giving me a chance to review the submitted paper.

Padkins and colleagues examined whether shock index (SI) was a predictor of in-hospital mortality among patients (n=9939) admitted to cardiac intensive care unit (CICU) at a single center (Mayo Clinic). The authors reported SI was associated with higher risk of death, with odds ratio of 1.07. They also found that both heart rate (HR) and systolic blood pressure (SBP) served as predictor of in-hospital mortality. These utilities were universal across various diseases/conditions.

I have some minor comments, listed below.

Major comment

None.

Minor comments

1. Abstract: As this is non-speciality journal, spell out CICU and BPM.

2. Methods (ln. #122-123): Why did the authors use “admission vital signs, clinical measurements, and laboratory values” at or after CICU admission? Prediction being the primary objective, it seems more common to use values before the event occurs.

3. Methods (ln. #143-144): Was the outcome fully ascertained in this notification system?

4. Results: Were there any patients with a device implemented for arrhythmia?

5. Results ((ln. #166)): Vasoactive agents were used in 7% of patient, but this figure was lower than those presented in Table 1 and 2.

6. Results (ln. #180-182): OR of 1.14 and 0.84 do not appear “strong” (at least to me).

7. Results (ln. #202): ? BPM/mmHg.

8. Results (ln. #205): Spell out “MCS”.

9. Discussion (ln. #307-310): Redundant as this was mentioned in Methods section.

10. Conclusion (ln. #317): “stronger” as compared what?

(End of comments)

Reviewer #3: The authors have performed an interesting study analyzing the association of the Shock Index with survival in a high volume CCU. The study is well designed. I have a few minor comments:

1. More description about the multivariate model would be useful

2. The adjusted data are mentioned in the results section, but seem like they would be helpful to include in the main figures of the paper as well.

6. PLOS authors have the option to publish the peer review history of their article (what does this mean?). If published, this will include your full peer review and any attached files.

Reviewer #1: No

Reviewer #2: No

Reviewer #3: No

---

## [Author Response · Author response to Decision Letter 0]

5 Jan 2024

Thank you for the interest in this original research manuscript. We believe this manuscript will be of interest to the audience of PLOS ONE based on the simplicity of using the shock index as an early risk stratification and prognostication tool in the cardiac intensive care unit. 

 We appreciate the thoughtful comments of the Reviewers and have addressed them in bold point-by-point format below in our response letter. We have made many adjustments and improvements to the manuscript to conform to PLOS ONE’s style requirements and to enhance the clarity of the manuscript. The suggested changes have certainly strengthened the impact of our manuscript and we hope you will find it suitable for publication in its revised format.

 Please do not hesitate to reach out with any additional questions or concerns.

---

## [Decision Letter · Decision Letter 1]

11 Jan 2024

PONE-D-23-31759R1Association between the Shock Index on Admission and In-Hospital Mortality in the Cardiac Intensive Care UnitPLOS ONE

Dear Dr. Padkins,

Thank you for submitting your manuscript to PLOS ONE. After careful consideration, we feel that it has merit but does not fully meet PLOS ONE’s publication criteria as it currently stands. Therefore, we invite you to submit a revised version of the manuscript that addresses the points raised during the review process.

We look forward to receiving your revised manuscript.

Kind regards,

Gaetano Santulli, MD, PhD, FAHA

Academic Editor

PLOS ONE

Journal Requirements:

Reviewers' comments:

Reviewer's Responses to Questions

**Comments to the Author**

1. If the authors have adequately addressed your comments raised in a previous round of review and you feel that this manuscript is now acceptable for publication, you may indicate that here to bypass the “Comments to the Author” section, enter your conflict of interest statement in the “Confidential to Editor” section, and submit your "Accept" recommendation.

Reviewer #2: All comments have been addressed

Reviewer #4: (No Response)

2. Is the manuscript technically sound, and do the data support the conclusions?

Reviewer #2: Yes

Reviewer #4: Yes

3. Has the statistical analysis been performed appropriately and rigorously? 

Reviewer #2: Yes

Reviewer #4: Yes

4. Have the authors made all data underlying the findings in their manuscript fully available?

Reviewer #2: Yes

Reviewer #4: (No Response)

5. Is the manuscript presented in an intelligible fashion and written in standard English?

Reviewer #2: Yes

Reviewer #4: (No Response)

6. Review Comments to the Author

Reviewer #2: Thank you for the satisfactory responses to my comments. I believe that the revised manuscript is ready for the publication.

Reviewer #4: The study aimed to assess the utility of the shock index (SI) in predicting in-hospital mortality among unselected patients admitted to a cardiac intensive care unit (CICU). Analyzing data from 9,939 patients admitted to the Mayo Clinic between 2007 and 2015, patients were categorized based on admission SI levels. Results revealed that elevated heart rate (HR) and decreased systolic blood pressure (SBP) were independently associated with higher in-hospital mortality after adjustment. Additionally, an incremental increase in SI correlated with a higher risk of in-hospital mortality, irrespective of the admission diagnosis. The study concludes that SI, a readily available and simple marker, can effectively predict in-hospital mortality in CICU patients at the time of admission.

The discussion should be implemented, including the following reports:

Relationship Analysis of Central Venous-to-arterial Carbon Dioxide Difference and Cardiac Index for Septic Shock.

The Shock Index revisited - a fast guide to transfusion requirement? A retrospective analysis on 21,853 patients derived from the TraumaRegister DGU.

Risk Score to Predict Need for Intensive Care in Initially Hemodynamically Stable Adults With Non-ST-Segment-Elevation Myocardial Infarction.

Iconography should be substantially improved (professional assistance should be sought and proof of such assistance should be provided).

The strengths and limitations of the study should be better addressed, taking into account all sources of potential bias or imprecision: Discuss both direction and magnitude of any potential bias.

7. PLOS authors have the option to publish the peer review history of their article (what does this mean?). If published, this will include your full peer review and any attached files.

Reviewer #2: No

Reviewer #4: No

---

## [Author Response · Author response to Decision Letter 1]

20 Jan 2024

Journal Requirements

Author’s Response: The reference list was reviewed and updated appropriately. All references are adequately cited and appropriate. No references have been retracted to the best of our knowledge. 

Reviewer Comments to the Author

Reviewer #4 

1. The study aimed to assess the utility of the shock index (SI) in predicting in-hospital mortality among unselected patients admitted to a cardiac intensive care unit (CICU). Analyzing data from 9,939 patients admitted to the Mayo Clinic between 2007 and 2015, patients were categorized based on admission SI levels. Results revealed that elevated heart rate (HR) and decreased systolic blood pressure (SBP) were independently associated with higher in-hospital mortality after adjustment. Additionally, an incremental increase in SI correlated with a higher risk of in-hospital mortality, irrespective of the admission diagnosis. The study concludes that SI, a readily available and simple marker, can effectively predict in-hospital mortality in CICU patients at the time of admission.

a. The discussion should be implemented, including the following reports:

Relationship Analysis of Central Venous-to-arterial Carbon Dioxide Difference and Cardiac Index for Septic Shock.

The Shock Index revisited - a fast guide to transfusion requirement? A retrospective analysis on 21,853 patients derived from the TraumaRegister DGU.

Risk Score to Predict Need for Intensive Care in Initially Hemodynamically Stable Adults With Non-ST-Segment-Elevation Myocardial Infarction.

Author’s Response:

We thank the Reviewer for pointing out these articles that we had missed during our literature review. We agree completely that the third of these is directly applicable to our work, and we have included a commentary regarding this important article in our updated discussion. The second article is somewhat relevant, but is in a very different population (trauma patients) looking at a different outcome (need for transfusion in hemorrhage)—we have mentioned this in our updated discussion. With due respect, we are having difficulty applying the findings of the first of these articles to our manuscript’s results, as it focused on a different aspect of physiology (arterial-venous CO2 differences) in a different population (septic shock); accordingly, we would prefer not to discuss this important but conceptually different work. 

Manuscript Excerpt:

Further, the ACTION ICU score is another scoring system that utilized HR and SBP and can be used in non-ST segment elevation myocardial infarction patients to predict ICU level of care needs (41). SI has also been studied in hypovolemic trauma patients and during hemodynamic resuscitation and has been found to predict mortality in these groups as well (42). All of these studies integrate the HR and SBP to predict risk and prognosticate outcomes, and thus, SI has been shown to predict prognosis and mortality in many patient populations especially cardiac patients. 

2. Iconography should be substantially improved (professional assistance should be sought and proof of such assistance should be provided).

Author’s Response:

We are uncertain what the Reviewer is referring to here, but we presume this is a comment regarding the quality of our figures. We do not have access to professional assistance with the figures, and this is a concern that was not previously raised. Accordingly, we would prefer not to make any dramatic changes to our figures without specific suggestions. We defer to the Editor in this regard, and are happy to make any changes that they feel are necessary to improve the presentation of our figures.

3. The strengths and limitations of the study should be better addressed, taking into account all sources of potential bias or imprecision: Discuss both direction and magnitude of any potential bias.

Author’s Response: 

We thank the Reviewer for this helpful comment. We agree that this analysis, like other retrospective cohort studies, has important limitations that must be understood to appreciate its implications. We have amended and expanded the Strengths and Limitations section, as shown below.

Manuscript Excerpt:

Strengths and limitations

 The primary strength of this analysis is the inclusion of a large cohort (nearly 10,000 unique patients) of CICU patients spanning nearly a decade (2007 to 2015), with extensive data regarding their hospitalization to provide deep characterization of the population. We purposefully included a heterogeneous group patients including the spectrum of illness severity from those with relatively uncomplicated acute cardiovascular disease to those with severe critical illness. This allowed us to examine the performance of the admission SI as a simple marker for risk stratification in patients with different disease processes and degrees of circulatory failure. 

Despite these strengths, this study has inherent limitations to single-center retrospective cohort analyses and prevents drawing causal inferences. A key source of potential bias is selection bias, as this single-center study cohort only included those admitted to the CICU at our facility and may not apply equally to patients who are not candidates for CICU admission for a variety of possible reasons. The CICU population at Mayo Clinic may differ from other populations in terms of baseline demographics, case mix, and resource utilization. Further, this CICU cohort at Mayo Clinic was relatively unselected, and these results may not apply equally to all subsets of the CICU patient population. Residual confounding is always possible in studies such as this one, and in this case an important potential source of residual confounding is overall severity of illness, which increased progressively as the SI increased and might be incompletely captured with the covariates we use for adjustment. In addition, our analyses focused only on initial vital signs, and it is likely that the evolution of changes in vital signs (e.g., SI) over time could provide enhanced prognostic value; a few patients did not have vital signs documented and were excluded. Further studies would be necessary to determine how a changing SI trajectory over time during a patient’s CICU and hospital stay affected mortality risk. Data regarding medication utilization prior to CICU admission (e.g., beta-blockers, anti-arrhythmics, and other antihypertensives) was unavailable and may influence the observed admission HR, SBP, and SI. Notably, other versions of the SI can be calculated, including the modified SI based on the mean arterial pressure and the diastolic SI based on the diastolic blood pressure. We found that the area under the curve value for discrimination of in-hospital mortality was equivalent among these permutations (0.66) and trivially higher for the standard SI (data not shown). We did not have data regarding the presence of implantable electronic cardiac rhythm devices or anti-arrhythmic drugs, which could have influenced the HR and thus the SI and its association with mortality. Finally, we could not determine each patient’s cardiac rhythm when the SI was calculated, preventing us from determining the effects of tachyarrhythmias and bradyarrhythmias on the prognostic associations of the SI. This is an important potential source of imprecision, as a variety of factors (including antiarrhythmic and vasoactive drugs as well as cardiac rhythm) can independently affect the SI and therefore affect the relationship between SI and outcomes by either weakening this association (i.e., if a change in SI occurred due to a benign condition that did not affect mortality) or artificially augmenting this association (i.e., if an increase in SI occurred due to a harmful condition that worsened outcomes). The true range of SI values within the cohort was very wide, with only a few individuals at the extreme values. Accordingly, we chose to categorize SI values into groups to ensure adequate group size for analysis, but such categorization results in information loss and our estimates are imprecise at the extremes of SI where there were sparse data; this could explain why we did not see dramatic incremental increases in mortality at the highest SI values (i.e., a plateau effect was present to some extent above a SI value of 1).

---

## [Editor Report · Decision Letter 2]

23 Jan 2024

Association between the Shock Index on Admission and In-Hospital Mortality in the Cardiac Intensive Care Unit

PONE-D-23-31759R2

Dear Dr. Padkins,

We’re pleased to inform you that your manuscript has been judged scientifically suitable for publication and will be formally accepted for publication once it meets all outstanding technical requirements.

Kind regards,

Gaetano Santulli, MD, FAHA

Academic Editor

PLOS ONE

---

## [Editor Report · Acceptance letter]

28 Feb 2024

PONE-D-23-31759R2 

PLOS ONE

Dear Dr. Padkins, 

I'm pleased to inform you that your manuscript has been deemed suitable for publication in PLOS ONE. Congratulations! Your manuscript is now being handed over to our production team.

Kind regards, 

on behalf of

Professor Gaetano Santulli 

Academic Editor

PLOS ONE